# Physicochemical and Sensory Assessments in Spain and United States of PGI-Certified *Ternera de Navarra* vs. *Certified Angus Beef*

**DOI:** 10.3390/foods10071474

**Published:** 2021-06-25

**Authors:** María José Beriain, María T. Murillo-Arbizu, Kizkitza Insausti, Francisco C. Ibañez, Christine Leick Cord, Tom R. Carr

**Affiliations:** 1Institute of Innovation & Sustainable Development in the Food Chain (IS-FOOD), Arrosadia Campus, Public University of Navarre (UPNA), Jerónimo de Ayanz Building, 31006 Pamplona, Spain; mariateresa.murillo@unavarra.es (M.T.M.-A.); kizkitza.insausti@unavarra.es (K.I.); pi@unavarra.es (F.C.I.); 2Nestle Purina PetCare Company, 1 Checkerboard Square, St. Louis, MO 63164, USA; christine.cord@purina.nestle.com; 3Department of Animal Science, University of Illinois at Urbana-Champaign, Urbana, IL 61801, USA; trcarr1@illinois.edu

**Keywords:** *Pirenaica*, Protected Geographical Indication, *Ternera de Navarra*, *Certified Angus Beef*, country of origin, USDA standard, sensory profile

## Abstract

The physicochemical and sensory differences between the PGI-Certified *Ternera de Navarra* (*CTNA*) (Spanish origin) and *Certified Angus Beef* (*CAB*) (US origin) were assessed in Spain and the USA. To characterize the carcasses, the ribeye areas (REAs), and marbling levels were assessed in both testing places. Twenty striploins per certified beef program were used as study samples. For sensory analysis, the striploins were vacuum packaged and aged for 7 days at 4 °C and 85% RH in each corresponding laboratory. Thereafter, the samples were half cut and frozen. One of the halves was shipped to the other counterpart-testing place. The fat and moisture percentage content, Warner Bratzler Shear Force (WBSF), and total and soluble collagen were tested for all the samples. The *CAB* carcasses had smaller REAs (*p* < 0.0001) and exhibited higher marbling levels (*p* < 0.0001). The *CAB* striploins had a higher fat content (*p* < 0.0001) and required lower WBSF (*p* < 0.05) than the *CTNA* samples. Trained panelists rated the *CAB* samples as juicer (*p* < 0.001), more tender/less tough (*p* < 0.0001), and more flavorful (*p* < 0.0001) than the *CTNA* counterparts. This study shows that beef from both countries had medium-high tenderness, juiciness, and beef flavor scores and very low off-flavor scores. Relevant differences found between the ratings assigned by the Spanish and the US panelists suggest training differences, or difficulties encountered in using the appropriate terminology for defining each sensory attribute. Furthermore, the lack of product knowledge (i.e., consumption habits) may have been another reason for such differences, despite the blind sensory evaluation.

## 1. Introduction

Globally, meat consumption demand is expected to rise in the coming years because of the population growth and rising incomes in developing countries [1]. According to the United Nations Food and Agriculture Organization, beef ranks third in the world’s most consumed meats [2]. The understanding of meat quality factors and consumers preferences is crucial to bridge the gap between the quality approach objectives in the abattoirs, the product physicochemical approach, and the sensory assessment.

Food quality is a complex and multidimensional concept and consumers’ quality expectations may not align with the definitions of food producers and academicians. Moreover, consumption habits and preferences may clearly influence the score assigned to meat coming from different origins. Hence, the country of origin and production practices can affect both meat quality and consumer acceptance [3,4]. As reported by Morales et al. [5], low- or trace-marbled beef was acceptable for many Chilean consumers, whereas consumers in Japan and Korea tend to prefer beef with abundant, evenly distributed marbling [6].

Marketing is about differentiation. Provenance and tradition serve to differentiate autochthonous, regional native products. In Europe, many food specialties are recognized by the name of the region, a term that involves the totality of natural factors (environmental conditions), and the activity of its inhabitants (“local know how”) that altogether determine the quality of the product [7]. These authors point out that Protected Geographical Indications (PGI) offer a guarantee to consumers, defining the conditions, the procedures, and the extent of protection, and a safeguard to protect the names of regional products. In Spain, the certified PGI-Certified *Ternera de Navarra* (*CTNA*) is designated for the veal produced in the autonomous community and province of Navarre (Northern Spain) under specifications surveilled by a local Regulatory Council [8] and has been the subject of several characterization studies [9,10]. On the other hand, in the United States, beef or veal does not bear a PGI quality seal but is subjected to other marketing strategies that include Certified Beef Programs under specifications surveilled by the Agricultural Marketing Service of the US Department of Agriculture (AMS-USDA) [11]. Such is the case of *Certified Angus Beef* (*CAB*), a pioneering branding program with wide international visibility based on the Angus breed/breed type [11,12]. Appendix A (Appendix A) depicts the main differences in minimum specifications for the respective certification of the *CAB* and CTNA programs.

The *Pireniaca* breed comprises 90% of the *CTNA* [13]. It is a meat-purpose breed that grazes in mountain areas, taking advantage of pastures mainly composed of native vegetation (graminoids stems, *Bachypodium pinnatum* leaves, *Festuca rubra* leaves, other graminoids, and forbs plus browse) [14]. Fattening is carried out in the same breeding farms, and feed diets are generally based on concentrate and straw. Typical young carcasses are obtained from young bullocks or heifers slaughtered at about 11–13 months of age.

Spanish cattlemen are currently looking to expand the local beef markets, including international markets. This strategy must consider the sensory evaluation and physiochemical traits of the final product to better predict the eating quality as it is perceived by the consumers in the target countries.

The United States’ beef production system and management practices are very diverse, and ongoing regional assessments have been conducted in order to guide the development of representative production systems [8,9,10]. Male castration is commonly practiced; therefore, castrated males (steers) prevail. In the USA, beef quality grades are determined on combinations of carcass maturity and marbling levels [15], and typical young carcasses are derived from steers or heifers rather than bullocks. Steers and heifers are typically fed with grain-based diets and slaughtered at about 15–28 months of age.

The association of marbling degree with juiciness and the tenderness of the meat is well established. Frank et al. [16] showed positive associations between intramuscular fat and overall flavor as well as juiciness and tenderness for Australian beef. Fat makes the matrix soft and easier to chew, therefore influencing the meat’s tenderness [17,18,19].

Initially, consumer’s choices are decided by basic (visual) perceptions on meat at the sale point, such as marbling or color. Palatability features are perceived upon tasting and beef overall liking is primarily defined by texture and flavor [19]. In a consumer study performed in Chicago and San Francisco, domestic beef was rated as more acceptable than its Argentinean counterpart [20]. In this survey, US beef was scored with higher ratings for juiciness, tenderness, flavor, and overall acceptability. Sitz et al. [18] reported that consumers from Denver and Chicago preferred domestic beef steaks versus those coming from Canada and Australia and suggested that the more desirable sensory ratings for the domestic beef may be due to the familiar taste of this type of meat. On the contrary, in a consumer sensory acceptance study performed in Spain by Beriain et al. [9], USDA prime beef (*Longissimus dorsi*) scored better than Spanish beef, the former one had a higher fat content and exhibited more desirable ratings for tenderness, juiciness, and flavor.

The meat industry devotes a lot of efforts to avoid the high variability in beef sensory quality. For the most part, the inherent variability detected in beef eating quality depends on the muscle physical and chemical properties, but a high level of palatability differences still remains due to the extrinsic characteristics of beef [21]. Therefore, consumers’ surveys are useful when assessing consumer’s preferences or willingness to pay. However, they are probably not the best option to compare the sensory attributes of meats from different sources or origins that should be evaluated by experienced panelists. There are few papers comparing the evaluation of certified beef carried out by a panel of experts from different countries in terms of meat sensory characteristics.

The objectives of this study were (a) to assess the differences in the physicochemical traits of samples derived from two different certified beef programs; these are, the PGI-Certified *Ternera de Navarra* (*CTNA*) in Spain and the *Certified Angus Beef* (*CAB*) in the United States; and (b) to assess the effect of a certified beef program on beef/veal sensory attributes as evaluated by two, separately trained, descriptive panels in Spain and the USA.

## 2. Materials and Methods

In Spain, the experimental part was performed at the Public University of Navarre (UPNA), whereas the experimental part in the USA was performed at the Meat Science Laboratory, University of Illinois at Urbana-Champaign (UIUC).

### 2.1. Animals Handling and Sampling

This study was performed by testing a total of 40 meat samples derived from two different biological types of cattle, as follows: 20 striploins derived from *CTNA* (based on *Pirenaica*, a large-framed, slow-maturing, continental breed)*,* and 20 *CAB* striploins imported from the USA, presumably derived from small-to-medium framed, earlier maturing Angus-influenced fed cattle.

The twenty yearling *CTNA* bullocks (507 ± 51 kg of BW and 366 ± 23 days of age) were born, raised, and harvested in Navarra. Animals were fed with commercial concentrate and straw. The handling experimental procedures of the *CTNA* cattle followed the European Directive 2010/63/EU, regulated by the Real Decreto 348/2000 in Spain. CNTA bullocks were slaughtered at a commercial abattoir in Pamplona (Navarre, Spain), in accordance with the Council Regulation for the protection of animals at slaughter [22]. Carcasses were chilled for 24 h in a conventional chamber at 2 °C (98% RH). The *Longisimus dorsi lumborum* (LDL) muscle was removed from each carcass from the first to the sixth lumbar vertebrae. Upon reception at the UPNA from the abattoir, the loins were vacuum packaged and aged for 7 days in the dark in controlled chambers at 85% RH and 4 °C, that had circulated air with one renovation every 24 h. Thereafter, the loins were half cut, and one portion was frozen at −20 °C and then shipped to UIUC. The other portion was cut into approximately 2.5-centimeter-thick striploins, vacuum packaged, and frozen at −20 °C until subsequent analysis. All samples were frozen by directly placing them in a freezer with a set temperature of −20 °C. Similarly, upon reception of the half part of the loins, the UIUC personnel cut the frozen loins into 2.5-centimeter-thick striploins using a band saw, the steaks were vacuum packaged, and kept frozen until the sensory analysis was performed.

The twenty *CAB* samples fulfilled the specifications at Tyson Fresh meats (Joslin, IL., USA) [23], 370 kg. After chilling for 24 h at 2 °C, loins were removed from the carcasses and shipped to the UIUC facilities under refrigeration (5 °C). Once there, loins were vacuum packaged and aged for 7 days at 4 °C and 95% RH. Afterwards, loins were cut into 2.54-centimeter-thick striploins and trimmed to a maximum of 0.6 cm subcutaneous fat thickness, vacuum packaged, and kept frozen until further sensory analysis was performed. At the same time, half of the loins were shipped in a frozen state to UPNA, where, upon arrival, they were cut into 2.54-centimeter-thick striploins, vacuum packaged, and kept frozen until further analysis was performed (Figure 1).

Either for physicochemical analysis or for sensory assessment in any location, the samples were thawed for 24 h at 4 °C prior to testing and were immediately analyzed.

### 2.2. Marbling Score and Longissimus Dorsi Area

In both countries these measurements were performed on the hanging carcass at the respective harvesting plant. The carcass evaluations performed followed the official guidelines of the EU beef classification system [24] and the USDA grading standards [15], in each respective country.

The marbling (visible intramuscular fat) was considered as the intermingling or dispersion of fat within the lean muscle. Graders evaluated the amount and distribution of marbling in the ribeye muscle at the cut surface after the carcass had been ribbed between the 12th and 13th ribs. For their evaluation, marbling scores were divided into 100 subunits. In general, however, marbling scores were divided in tenths within each degree of marbling (e.g., Slight 90, Small 00, Small 10, etc.).

The REA was measured at the last rib. It was measured with an USDA plastic grid of equally spaced dots representing a scale of 0.654 cm^2^. The interior dots that were completely within the perimeter of the ribeye muscle were counted. The boundary dots that were on the perimeter of the ribeye muscle were also counted, but the calculated vale was divided by two and the results added to the previous interior dot number.

### 2.3. Warner Bratzler Shear Force

Steaks were thawed at 4 °C overnight prior to shear force evaluation. Steaks were cooked on a 180 °C preheated grill, turning the steak every 4 min until an internal temperature of 70 °C was reached on an open heart-grill (GR3000, JATA S.A, Tudela, Spain). The final temperature was monitored using copper-constantan thermocouples (Data Acquisition System was a portable digital thermometer Hanna in Spain). After cooking, steaks were cooled for 2 h at 5 °C and then, seven cores with a cross section of 1 cm × 1 cm × 3 cm were cut from each steak parallel to the longitudinal axis of the muscle fiber’s orientation. Cores that were not uniform in size, had obvious connective tissue defects, or were not representative of the sample were discarded. Shear tests were performed with a TA-XT2i texture analyzer (Stable Micro System Ltd., Surrey, UK). The equipment was fitted with a 30-kilogram load cell and a Warner–Bratzler with a V slot shear attachment, fulfilling the requirements for this type of test [25]. A 5-kilogram weight was used for system force calibration. Test speed of the shear was set at 10 mm/s. Peak shear force values for the seven cores were averaged for each sample. The measurement of the maximum force (kg) as a function of knife movement (mm) and the compression to shear a sample of meat was considered as the hardness (toughness) of meat. Data were collected via Texture Expert v.2.0. software (Stable Micro Systems Ltd., Surrey, UK).

### 2.4. Proximate Analysis

The following chemical constituents were determined on thawed samples. In UIUC laboratories, moisture contents were determined using the ISO standard [26]. One steak from each striploin was trimmed of surrounding adipose and connective tissues and homogenized individually using a Cusinart Food Processor (Model DLC 5-TX, Cuisinart, Stamford, CT, USA). Duplicate 10-gram samples of each homogenized steak were weighed, placed in an aluminum film, and covered with filter paper. Each sample was oven dried (110 °C for 48 h) and weighed to determine moisture content. The additional homogenized tissue was frozen at −20 °C to be used for collagen quantification.

For the determination of intramuscular fat content in UPNA, LDL steaks were thawed, trimmed of connective and surrounding adipose tissues, and cut into small pieces for grinding (grinder Moulinex 800W Dpa 251, Groupe Seb Iberica Ltd., Barcelona, Spain). The fat analysis was performed in duplicate with the use of heat and solvents, which is derived from the Soxhlet type extraction [27]. Briefly, 6 g of ground meat was digested with boiling HCl 3 N for one hour. For that, a heating plate (Combiplac, JP Selecta, Barcelona, Spain) was used. Then, samples were filtered through filter paper (Albet 242Ø), and the filter with the fat dried at 70 °C for at least 12 h. After that, fat was extracted by using a Soxhlet system with ethyl ether as the solvent. The fat content was measured by the gravimetric differences of the round bottom flask that collected the extraction solvent during the Soxhlet cycles and correlated with the starting weighted meat quantity. The results were expressed as a percentage of fresh meat.

### 2.5. Total Collagen Quantification

The total collagen determination in US laboratories was conducted using a method based on AOAC [28] with modifications for a microplate assay. Duplicate 4-gram samples were hydrolyzed in 30 mL of sulfuric acid for 36 h at 105 °C. Hydrolysates were brought to a total volume of 100 mL with water and filtered. Hydroxyproline quantitation was performed in a deep 96-well plate, where 50 μL of each hydrolysate sample was added to 750 μL of water. Then, an oxidant solution (400 μL of cloramine T) was added to each well, and the plate was incubated at room temperature for 20 min. Afterwards, a color reagent (400 μL of Ehrlich’s/pDMAB solution) was added to each well, and the plate was heated in a water bath at 60 °C for 15 min. Samples were read on a plate reader against a hydroxyproline standard curve at 557 nm to determine the hydroxyproline concentration in the sample, then multiplied by 8 to calculate the percent collagen in the sample.

### 2.6. Soluble Collagen Quantification

In UPNA laboratories, the soluble collagen content was obtained using the method described by Hill [29]. Four grams of meat sample were dissolved in 1/4-strength Ringer’s solution in a 77 °C water bath for 70 min. After that, the sample was centrifuged, and the upper phase of each sample was hydrolyzed with HCl (50%) by boiling in a reflux for 7 h [30]. Then, the pH was adjusted in a range between 6–7, and the volume was brought to 100 mL with distilled water. The hydrolyzed solution was filtered by gravity through an Albert filter and kept at 5 °C until the hydroxyproline content determination. Hydroxyproline quantitation was performed in a glass tube where 1000 μL of each hydrolysate sample was added. Then, 500 μL of oxidant solution (cloramine T prepared in sodium acetate and isopropanol) was added to each tube, and they were incubated at room temperature for 20 min. Thereafter, 500 μL of the color reagent (Ehrlich’s/pDMAB prepared in isopropanol and percloric acid) was added to each tube and heated in a water bath at 65 °C for 15 min. Samples were read on a UV spectrophotometer against a hydroxyproline standard curve (0–14.4 ppm) at 550 nm to determine the hydroxyproline concentration in the sample, then multiplied by 7.52 [31] to calculate the percent of soluble collagen in the samples.

### 2.7. Sensory Assessment

As described in Section 2.2, a total of 40 striploins steaks were procured either in the USA (*CAB*) or Spain (*CTNA*) and were sensorially tested in both research centers (UPNA in Spain and UIUC in the USA) by six panelists. In the United States, the training, cookery, and sensory evaluation followed the guidelines of the American Meat Science Association [32].

The rigorous methodology of Gorraiz et al. [33] was used in UPNA to train the descriptive panels and to select the attributes to evaluate the meat’s sensory quality. In UPNA, panelists were highly trained in beef tasting with an average of 10 years’ experience. All were rewarded with a voucher for buying beef.

Steaks were thawed overnight at 4 °C prior to each sensory evaluation session. Steaks were cooked in 180 °C preheated open-hearth electric grills (model 450 Farberware Cookware, Fairfield, CA, USA used at UIUC; and model GR3000, JATA S.A, Tudela, Spain used at UPNA) inside aluminum foil, turning the steak every 4 minutes until an internal temperature of 70 °C was reached. Every steak was then trimmed of any external connective tissue, cut into 3.5 cm × 1 cm × 1 cm rectangle samples, wrapped in coded aluminum foil and kept in a warmer, provided with sand, until the analysis was performed. Samples were coded with a randomized 3-digit number, and they were presented in a random order to the panelist in order to avoid the carry-over effect. Seven tasting sessions were conducted to obtain at least six judgments per steak. Briefly, in each tasting session 3 *CTNA* and 3 *CAB* samples were presented to each judge, except for the first session when two samples per certified beef were tasted.

Panelists were seated in individual booths under red lighting and were provided with water to cleanse the palate. Panelists evaluated samples for tenderness, juiciness, and beef flavor as described by Gorraiz et al. [33] on a 15-centimeter unstructured line scale with an anchor in the middle of the scale where 0 = extremely tough, extremely dry, no beef flavor, and no off-flavor and 15 = extremely tender, extremely juicy, intense beef flavor, and intense off-flavor. In addition to these attributes, the Spanish panelists were asked to evaluate liver- and fat-like flavor, and the US panelists were asked to assess the perception of off-flavor.

### 2.8. Statistical Analysis

For the physicochemical characterization and sensory assessment, the experimental unit was the animal (*n* = 40), and the fixed effect was the certified beef program. Mean values were compared using the Student’s *t*-test for paired samples (within each certified beef program/country). Significant statistical differences were declared at *p* ≤ 0.05. The sensory quality of *CTNA* and *CAB* samples was also investigated using a discriminant analysis (test of equality of means of Lambda Wilks groups; *p* ≤ 0.05). All data were analyzed using IBM SPSS Statistics, version 25.0 for Windows (IBM Corporation, Armonk, NY, USA).

## 3. Results and Discussion

The descriptive statistics and mean comparisons for the marbling scores and REAs, physicochemical, and sensory attributes are presented in Table 1, Table 2 and Table 3, respectively. It was noted that the marbling scores showed a large standard deviation (SD) within each certified beef program. As this determination depends on the graders´ criteria and the carcass’ fatness, a high variability may be expected [34]. The REA variability was ca. 20% for the *CTNA* and ca. 7% for the *CAB* samples. The other analytical parameters, including the sensory traits, showed an SD lower than 10 percent.

### 3.1. Marbling Score and REA

The mean marbling score evaluated in both laboratories for the striploins of different experimental groups showed greater values for the *CAB* striploins than for the *CTNA* counterpart (Table 1), with highly significant differences (*p* < 0.0001). These results were expected due to the different genetic background and livestock practices performed in the two countries of origin. Additionally, because of the program’s specifications (Appendix A. Appendix A). The vast majority of US cattle are grain fed and for their carcasses to be branded as *CAB,* they must have a marbling score of modest or higher [11]. On the other hand, *CTNA* bullocks were subjected to shorter periods of grain feeding and the requirement for this type of certified beef is that the intramuscular fat percentage should be around 2%, which is usually achieved through the feeding protocol of the Protected Designation of Origin program [8].

The *CTNA* carcasses showed larger REAs than those of the *CAB* (*p* < 0.0001), with around 120 and 78 cm^2^, respectively. In general, the REA values are directly proportional to the carcass weights reported for the experimental groups, which averaged 330 and 370 kg for the *CTNA* and the *CAB* samples, respectively. The REA value of 106 cm^2^ was reported by Alberti et al. [35] for *Pirenaica* carcasses. REA area values ranging from 64.8 to 111.9 cm^2^ were reported by Nelson et al. [12] for Angus, Hereford–Angus cross, and Northern animals. The *CAB*’s REA value obtained in this work is within the above-mentioned range.

### 3.2. Warner–Bratzler Shear Force

WBSF showed lower mean values (Table 2) for the *CAB* striploins steaks as compared to those from *CTNA* (i.e., 3.86 vs. 5.90 kg). (*p* < 0.0001). Accordingly, the *CNTA* striploins were classified as “tough” according to a tenderness classification [30], as the averaged WBSF value was above 5.7 kg, whereas the *CAB* striploins were considered as intermediate in tenderness [36].

The mean WBSF values obtained for *Pirenaica* beef by Panea et al. [37] was 3.8 kg (*n* = 55); whereas, for *CAB*, the mean WBSF value reported by Nelson et al. [12] was 4.15 kg. Both precedent reports [12,37] support the values obtained in this study (5.9 and 3.86 kg for the *CTNA* and *CAB* samples, respectively). It can be claimed that breed had a strong impact on the WBSF values in accordance with the results obtained by López-Pedrouso et al. [38], who studied this textural trait in three different Spanish beef breeds.

### 3.3. Proximate Analysis

The mean fat content of the *CAB* samples was higher (*p* < 0.0001) than that of the CNTA striploins, which corresponds well with the statistical differences found in the marbling scores (Table 1). The mean values of the moisture content of the *CAB* striploins were about 10% lower (*p* < 0.0001; Table 1) than that of the *CNTA* counterparts. It is most likely that the explanation for these results is the inverse relationship between the fat and moisture contents in proximate composition; this is, as the fat content was higher in the CAB samples, there was a concurrent decrease in the moisture content [16,39,40].

### 3.4. Total and Soluble Collagen

The total collagen values did not vary with the certified beef program (*p* = 0.2485). The results for the *CTNA* samples are in accordance with those reported by Sañudo et al. [41] (3.5 and 0.27 mg/g for total and soluble collagen, respectively). The latter authors also evaluated these two parameters for other beef Spanish breeds and reported ranges for total collagen and soluble collagen of 4.1–2.8 and 0.27–0.54 mg/g, respectively. The values obtained in the present study were within those ranges [41].

A higher soluble collagen content typically indicates more tender meat [29], which was not supported by the WBSF values for the two certified beef programs. The possible transformation of soluble collagen into an insoluble collagen after meat cooling would explain the low values. Wheeler et al. [42] reported that WBSF was greater (indicating tougher meat) in steaks that exhibited lower marbling scores in *Bos taurus* cattle. The same study also reported a decreased variation in WBSF as the marbling score increased, which may explain the observed differences in the WBSF values obtained for each beef program.

### 3.5. Sensory Assessment

The sensory panelists in the USA described both types of striploins with medium-high scores for tenderness, juiciness, and beef flavor. Panelists found the *CAB* to be juicier (*p* = 0.0099), more tender (*p* < 0.0001), and more flavorful (*p* < 0.0001) than the *CTNA* samples (Table 3). A similar assessment was provided by the UPNA panel, which defined the *CAB* steaks as juicier (*p* = 0.0018), more tender (*p* < 0.0001), and more flavorful (*p* = 0.0004) than those of CTNA (Table 3). The marbling and the intramuscular fat percentage values obtained for the CAB samples may explain the higher ratings for juiciness because higher fat levels in meat produces a higher initial juiciness sensation in the mouth by increasing mouth lubrication and leading to a perceived improvement in tenderness [43]. It is a well-known relationship between marbling level and sensory attributes, including flavor perception, regardless of breed. In such a way, a higher intramuscular fat percentage is clearly linked to overall liking [43,44,45,46]. As fat is the primary driver of beef flavor and acceptability, the results presented herein support that, irrespective of the panelist’s country of origin; the *CAB* samples (containing 5.84% fat) were perceived as more tender and more desirable in beef flavor compared to the *CTNA* samples (containing only 0.85% fat).

In a sensory trial run in the USA, consumers rated corn-fed domestic beef higher in flavor, juiciness, and tenderness than Australian grass-fed beef, suggesting that the palatability characteristics of corn-fed beef were perceived as more desirable by US consumers [18]. The study reported fat levels of 8.82 and 6.12% for the US and Australian samples, respectively. A different study reported that US consumers did not detect differences in beef flavor when tasting meat from different breeds produced under similar livestock practices [42]. These findings, along with the present results, would suggest that the animal’s diet and the production practices had a greater impact on beef flavor than the marbling score. The beef flavor was detected with higher intensity in the CAB striploins than in the CNTA counterparts by trained panels in both countries. Similar sensory results in both testing places showed that panelists in the USA and in Spain were appropriately trained to detect differences between the samples and provided consistent assessments. The results obtained were in accordance with the assessments obtained from previous consumer studies, in which *CAB* received higher palatability scores as compared to non-US grass-fed beef [18,20,47].

Trained US panelists in the current study perceived less off-flavor in the *CAB* samples as compared to the *CTNA* counterparts (0.32 cm vs. 0.88 cm). The off-flavors detected for the *CAB* samples were described as “buttery” or “metallic”, whereas the off-flavors for the *CTNA* samples were described as “grassy”, “metallic”, “acidic”, “bloody”, and “serumy.” Although there was a significant difference in off-flavor incidence between the two certified beef samples, the mean off-flavor rating was less than 1 on a 15-point scale in any case, indicating a very low incidence of off-flavors in both types of certified beef.

Since the descriptors used in each testing place were slightly different due to differences in terminology and translation difficulties from English to Spanish, it was not possible to make a direct comparison between off-flavor and liver-like flavor, even though several panelists in the USA described the off-flavor in the *CTNA* samples as “livery” or “metallic”. These differences in the experiment may be partially attributed to the different tasting experiences and expectations of the panelists in these two different countries. Although both panels were highly trained, the US panelists may have been less sensitive to a liver-like flavor than the Spanish panelists. For US consumers, an intramuscular fat level below 3% is not acceptable [48]. As indicated by Arshad et al. [47], there are several factors influencing beef flavor, particularly fatty acids. The mineral composition and fatty acid profile affect the development of meat off-flavors, such as a liver-like flavor [49]. Another possible reason might be that the descriptors used in both sensory evaluations were not clear enough for panelists to accurately describe the differences between the samples. We must acknowledge this difficulty as a weakness of our work because the vocabulary should had been defined before testing to obtain a reliable and robust sensory assessment from both panels. Furthermore, despite their training, there may have been some inherent bias for panelists to prefer certain flavors more than others. Dransfield et al. [50] conducted trained sensory panels at five European locations using beef from eight different countries. These researchers reported that, even though the perceived differences in tenderness and juiciness were correlated among the panels, some variation was partly attributed to regional differences in reference to juiciness and tenderness [50]. A follow-up study [51], which used similar panelists for evaluating the effects of cattle breed and postmortem aging, reported similar findings. Panels in five European countries were able to distinguish differences in tenderness and juiciness fairly consistently, but there were differences in evaluations of beef flavor [51].

The values of the common descriptors to both panels (juiciness, tenderness, and beef flavor) were used in discriminant analysis. Two canonical functions were obtained and explained 99.8% of the global variability of data. The first function was associated with tenderness (negatively) and juiciness (positively). The second function was associated with beef flavor (positively). Function one could be defined as the “texture factor” and function two as the “flavor factor”. The results plotted in Figure 2 show that it is possible to differentiate the two types of meat with only three sensory attributes (juiciness, tenderness, and beef flavor). The *CAB* samples are in the upper quadrants. On the contrary, the *CTNA* samples are in the lower quadrants. The Spanish panel broadly differentiated both meats according to beef flavor. The two panels discriminated both meat types according to their beef texture.

## 4. Conclusions

Based on these results, US *Certified Angus Beef* striploins showed a clear advantage in textural quality as indicated by their lower WBSF and more desirable tenderness ratings, notwithstanding the similarities between CAB and CTNA in total and soluble collagen contents. The noticeable differences between the samples from both certified beef programs in the intramuscular fat and moisture percentages are explained by their inherent genetic and management backgrounds. The same applies to the larger REA and lower marbling scores of the *CTNA* striploins in comparison with those of *CAB*.

In terms of sensory quality, the panels in the two countries concur that the *CAB* striploins outperformed the *CTNA* samples in juiciness, tenderness, and flavor.

The relevant numerical differences found between the rating assigned by the Spanish and the US panelists suggest training differences, or difficulties encountered in using the appropriate terminology for defining each sensory attribute. Furthermore, the lack of product knowledge (i.e., consumption habits) may have been another reason for such differences, despite the blind sensory evaluation.

## Figures and Tables

**Figure 1 foods-10-01474-f001:**
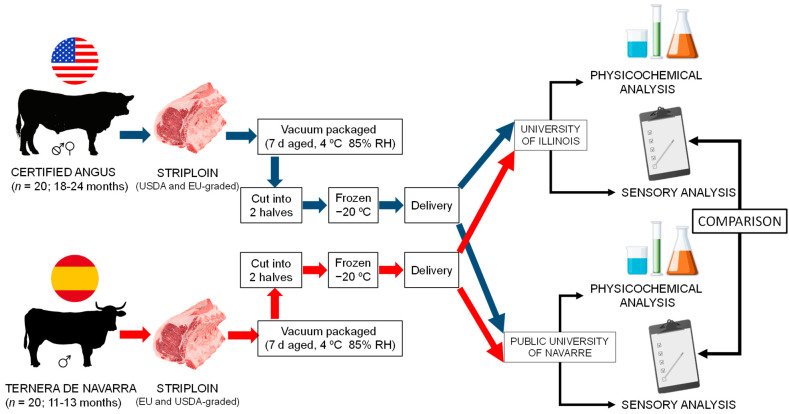
Illustration for experimental design of current study.

**Figure 2 foods-10-01474-f002:**
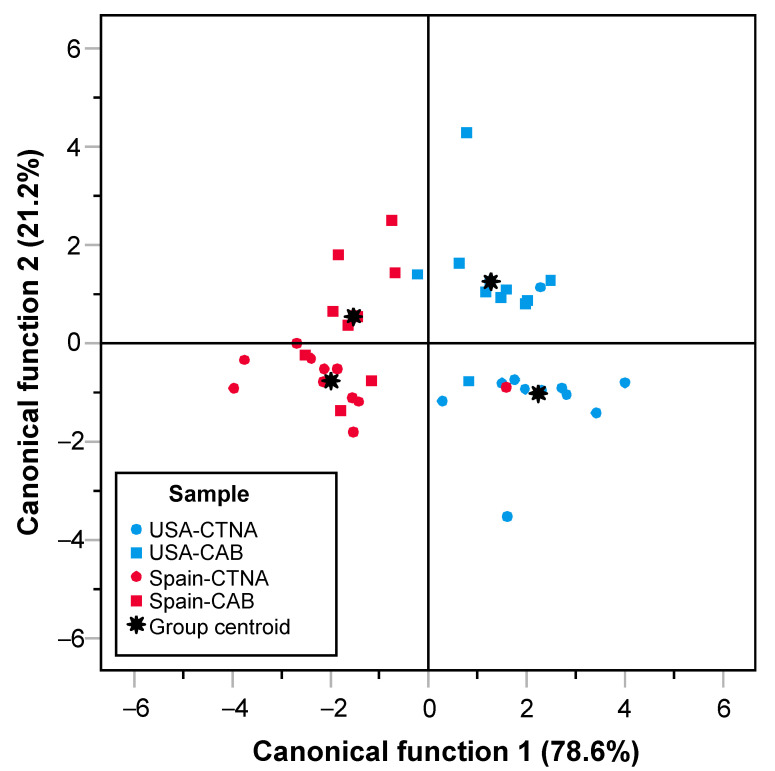
Spatial location of the PGI-Certified *Ternera de Navarra* (*CTNA*) and *Certified Angus Beef* (*CAB*) samples according to common attributes assessed in the USA (UIUC) and Spain (UPNA). Function 1 (texture factor) was associated with tenderness and juiciness, and function 2 (flavor factor) was associated with beef flavor.

**Table 1 foods-10-01474-t001:** Descriptive statistics and mean comparison for degree of marbling and *Longissimus dorsi thoracis* eye area (REA) according to EU classification and USDA grading systems for carcasses derived from bullocks PGI-Certified *Ternera de Navarra* (*CTNA*) and *Certified Angus Beef* (*CAB*) steers, respectively.

Attribute	*CTNA*				*CAB*				SEM	*p*-Value
	Mean	Min	Max	SD	Mean	Min	Max	SD		
Marbling ^1,3^	307.5 Traces^07^	230.0 Devoid^30^	410.0 Slight^10^	45	837.5 Slightly abundant^73^	730.0 Moderate^30^	1090.0 Abundant^90^	86	15.80	<0.0001
Marbling ^2,4^	102.5 Traces^07^	30.0 Practically devoid^30^	210.0 Slight^10^	50	739.0 Slightly abundant^75^	620.0 Moderate^30^	920.0 Abundant^90^	93	16.49	<0.0001
REA (cm^2^) ^1,3^	104.47	83.87	136.77	14.09	77.89	66.32	98.64	7.62	2.53	<0.0001
REA (cm^2^) ^2,4^	141.96	106.54	181.41	20.79	78.36	69.51	99.02	7.35	3.48	<0.0001

^1^ Tested in the USA; ^2^ Tested in Spain; ^3^ Following the USDA beef grading standards [15]; ^4^ Following the EU beef classification system [24].

**Table 2 foods-10-01474-t002:** Descriptive statistics and mean comparison for physicochemical traits of striploins derived from PGI-Certified *Ternera de Navarra* (*CTNA*) and *Certified Angus Beef* (*CAB*), respectively.

Attribute	*CTNA* Striploins				*CAB* Striploins				SEM	*p*-Value
	Mean	Min	Max	SD	Mean	Min	Max	SD		
Fat (%) ^2^	0.85	0.39	1.33	0.25	5.84	3.12	8.92	1.70	0.27	<0.0001
Moisture(%) ^1^	75.43	72.92	76.34	0.82	65.41	58.02	67.91	2.51	0.41	<0.0001
Total Collagen(mg/g) ^1^	2.52	1.45	3.31	0.48	2.69	2.09	3.33	0.40	0.10	0.2485
Soluble Collagen (mg/g) ^2^	0.41	0.27	0.78	0.15	0.43	0.24	0.71	0.11	0.29	0.6488
Shear Force (kg) ^2^	5.90	2.20	8.25	1.54	3.86	2.84	5.46	0.55	0.25	<0.0001

^1^ Tested in the USA; ^2^ Tested in Spain.

**Table 3 foods-10-01474-t003:** Descriptive statistics and mean comparison of sensory descriptive ratings for striploins derived from PGI-Certified *Ternera de Navarra* (*CTNA*) and *Certified Angus Beef* (*CAB*) as tested by trained panels in Spain and the USA, respectively.

Attribute	*CTNA* Striploins				*CAB* Striploins				SEM	*p*-Value
	Mean	Min	Max	SD	Mean	Min	Max	SD		
Tenderness ^1^	8.14	5.70	10.10	1.36	9.92	7.63	11.90	1.26	0.29	<0.0001
Juiciness ^1^	9.20	6.80	10.77	1.12	10.06	8.12	11.78	0.87	0.22	0.0099
Beef flavor ^1^	6.20	4.33	7.93	0.88	7.99	6.55	10.47	1.01	0.21	<0.0001
Off-flavor ^1^	0.88	0.17	1.75	0.50	0.32	0.07	0.78	0.22	0.09	<0.0001
Tenderness ^2^	8.72	5.46	9.72	0.86	9.86	7.77	11.90	0.96	0.22	<0.0001
Juiciness ^2^	6.05	3.66	9.19	1.34	7.39	5.69	9.48	1.19	0.28	0.0018
Beef flavor ^2^	6.25	3.77	7.69	1.00	7.28	5.90	8.70	0.77	0.18	0.0004
Liver-like flavor ^2^	3.77	2.63	5.04	0.75	3.23	2.12	5.38	0.79	0.17	0.0353
Fat flavor ^2^	2.83	1.77	3.66	0.57	4.27	2.80	6.86	1.01	0.18	<0.0001

^1^ Tested in the USA; ^2^ Tested in Spain.

## Data Availability

All data from the research conducted are available on request from the corresponding author.

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
