# Peer review of "Physicochemical and Sensory Assessments in Spain and United States of PGI-Certified *Ternera de Navarra* vs. *Certified Angus Beef"

_foods, 2021, doi:10.3390/foods10071474_

Round 1

Reviewer 1 Report

Current trends in food consumption indicate a search for new sources of protein, for example from edible insects or in vitro meat. Cultured meat may be a novel food that would overcome the

limitations of traditional meat production. However, is the concept of novel foods without flaws?

Scientists and consumers have many objections to the acceptance of novel foods. Therefore, the publication topics presented for review are still relevant.

The article entitled “Sensory Assessment of Pirenaica vs Certified Angus Beef by Two Trained Panels in Spain and USA” submitted for review presents good scientific value and can be published in Foods after improvements. Below are comments that should be referred to and possibly corrected in the text: 

  1. Was the pH of the meat measured before and after the maturation process ?. 
  2. Whether carrying out freezing of sliced and cut samples resulted in differences in meat quality. The method and time and freezing is not given. What was the freezing technique?. This is important in terms of meat quality after thawing. 
  3. Did both teams (US, Spain) evaluate samples after the same amount of time after freezing?. 
  4. In how many replicates were sensory tests conducted?. Whether samples for evaluations were coded individually and given in a random order to avoid the carry-over effect?. 
  5. In how many replicates were texture measurements performed? 
  6. There would be a added value to the study if objective tests were conducted and related to sensory evaluation (debatable point): 

- assessment of marbling using computer image analysis, 

- assessment of colour using CIE Lab and Chroma (C*) and hue angle (h*) (Analysis of chromatic components of colour, Chroma (C*) and hue angle (h*) allows more complete identification of consumer perception of meat colour). 

- assessment of post-mortem changes in meat - determination of myofibril fragmentation index (MFI). 

Author Response

Dear reviewer #1,

In this document, we address the comments and points raised, and we indicate the changes made in the manuscript (marked using Track Changes). The original comments made by the reviewer appear in italics and blue type, whereas our responses appear in normal case.

Comments and Suggestions for Authors 

Current trends in food consumption indicate a search for new sources of protein, for example from edible insects or in vitro meat. Cultured meat may be a novel food that would overcome the limitations of traditional meat production. However, is the concept of novel foods without flaws? 

Scientists and consumers have many objections to the acceptance of novel foods. Therefore, the publication topics presented for review are still relevant. 

The article entitled “Sensory Assessment of Pirenaica vs Certified Angus Beef by Two Trained Panels in Spain and USA” submitted for review presents good scientific value and can be published in Foods after improvements. Below are comments that should be referred to and possibly corrected in the text: 

Response: Thank you very much for the introduction made focusing on the topic of the manuscript. The authors appreciate the pertinent comments of Reviewer #1. The authors have incorporated all of them in the revised manuscript. 

  1. Was the pH of the meat measured before and after the maturation process?. 

Response: pH of the Longisimus dorsi muscle was measured in between the 6th and 12th ribs after the 24 hours of chilling but not during the aging process. No additional information about pH has been added to the manuscript. 

  1. Whether carrying out freezing of sliced and cut samples resulted in differences in meat quality. The method and time and freezing is not given. What was the freezing technique?. This is important in terms of meat quality after thawing.

Response: A sentence in line 114 has been added as follows: 

“All samples were frozen by directly placing them in a freezer with a set temperature of –20 ºC.” 

  1. Did both teams (US, Spain) evaluate samples after the same amount of time after freezing?. 

Response: Sentence in lines 126-127 has been nuance in the following way:  

“Either for analysis or for sensory assessment, the samples were thawed for 24 h at 4 °C, both in USA and in Spain, prior to testing and were immediately analyzed”. 

  1. In how many replicates were sensory tests conducted?. Whether samples for evaluations were coded individually and given in a random order to avoid the carry-over effect?.  

Response 1: A new sentence in section 2.7 has been added (L229-230). 

“Briefly, on each tasting session 3 Pirenaica and 3 CBA samples were presented to each judge, except for the first session where two samples per breed were tasted.” 

Response 2: Sentence in lines 226-228 has been nuance in the following way:  

“Samples were coded with a randomized 3-digit number, and they were presented in random order to the panelists in order to avoid the carry over effect.

For clarification the following figure is shown to the reviewer.

Figure: Distribution of meat portions for sensory evaluation. 

  1. In how many replicates were texture measurements performed?  

Response: L151-153. “After cooking, steaks were cooled for 2 h at 5 °C and then, seven cores with a cross section of 1x1x3 cm were cut from each steak parallel to the longitudinal axis of the muscle fiber orientation.”

  1. There would be a added value to the study if objective tests were conducted and related to sensory evaluation (debatable point):  

- assessment of marbling using computer image analysis,  

Response: Many thanks for the feedback. In future studies we will take it into consideration.  

- assessment of colour using CIE Lab and Chroma (C*) and hue angle (h*) (Analysis of chromatic components of colour, Chroma (C*) and hue angle (h*) allows more complete identification of consumer perception of meat colour).  

Response: The authors do agree with the comment. However, as this measurement was not performed at both laboratories, for it has not been included in the manuscript. 

- assessment of post-mortem changes in meat - determination of myofibril fragmentation index (MFI).  

Response: Many thanks for the feedback. In future studies we will take it into consideration.  

Additionally, the authors wish to indicate to the reviewer that due to the removal, modification or addition of some paragraph, the references have been updated. However, no change control for this action was kept due to the difficulties this action provided on reviewing the new version of the manuscript.

Reviewer 2 Report

Overall, this study has some merit, although the choice of using only 6 trained panellists gives me low confidence in the scientific judgement and experience of the authors. It is the absolute minimum number any self-respecting sensory scientist would use, and that would be just for screening purposes(HEYMANN, MACHADO, TORRI, & ROBINSON, 2012), https://www.ifst.org/sites/default/files/Sensory%20and%20Consumer%20Science-.pdf.

Why conduct an international study and put so few resources into it? I don’t get it! It is a serious potential flaw which needs explanation. Were the panellists highly trained in just tasting beef? How many years’ average of experience? How much training? Paid or unpaid? Much more information required, to convince me that 6 panellists was a wise choice!

In general, the English expression sounds very Spanish. Have the US co-authors had input into fixing up the English expression? I would say not. Have they even seen it? There is a lot of room for improvement! Please thoroughly revise the English using native speakers!

Abstract

Please be clear and unambiguous in your descriptions

L16 the trained panel assessed taste and texture so remove the word taste

L17 It is obvious they are already obtained… remove

L16 (Pirenaica, Spain) and (Certified Angus Beef, USA) … need to declare from which country

L18 I assume they were all frozen, not just the ones sent to the other country!

L19 visual marbling

L20 Indeed is not required

L22 greater à higher

L32 because of

L37-42 For example, they can mention the perception of marbling or fat in meat is considered quite differently in different countries (Damian Frank, Joo, & Warner, 2016; Lee, Yoon, & Choi, 2019)

L45 ‘ not required, these regulations establish..

L45-59 Much of this information is not necessary, please reduce

L66-77 They really need to describe the very strong relationships between marbling and flavour and tenderness. It is very well established and holds true regardless of country. (D. Frank, Ball, Hughes, Krishnamurthy, Piyasiri, Stark, et al., 2016; Damian Frank, Kaczmarska, Paterson, Piyasiri, & Warner, 2017; Joo, Hwang, & Frank, 2017)

Figure 1. Surely all the steak samples were frozen. Otherwise the tenderness and flavour will be totally different!

L214 Human ethics approval number in US and Spain?

L216 in both evaluation centres

L219 Grill temperature, cooking time?

L221 1 cm 3 is a very tiny amount to taste! 1 g of beef? How could you even chew it? Surely not!?

L223 In hot sand? What does that mean?

Are they saying that each panellist tasted 80 pieces of steak? Over 7 sessions? Please be clear how steaks from 40 Angus and 40 Pirenaica loins were allocated?

L244 Marbling scores had a large SD. It is not obvious what the word and superscript number means in Table 1. The whole point of a marbling score is a number. Please just use numbers in table. Interpretation of the numerical scores can be assisted by a table legend.

Tables: tested in USA, tested in Spain (not at)

Not sure why a discussion about SD is required in a separate section? Just discuss with data

L271 Pirenaica loins had larger average LEA than CAB à not showed (many other examples of this!)

274 Surely, they have the individual carcass weights for each animal. Why not present the actual correlations with a regression equation?

L288 were close to the ones obtained in this study

L293 Unless they performed a correlation (they have the data so why not?) please do not discus it. If you do please show r value, p value and regression equation and p value.

9-fold

L297 Please cite other work like (D. Frank, et al., 2016; Thompson, 2004; Thompson, Polkinghorne, Hwang, Gee, Cho, Park, et al., 2008) that also show this well-known relationship between fat and moisture. There are many studies showing this.

L313 Why state UIUC? Just Say the US sensory panellists

Please discuss the well-known relationship between marbling level and sensory attributes, including flavour release regardless of breed (Corbin, O'Quinn, Garmyn, Legako, Hunt, Dinh, et al., 2015; Damian Frank, Kaczmarska, Paterson, Piyasiri, & Warner, 2017; Hunt, Garmyn, O'Quinn, Corbin, Legako, Rathmann, et al., 2014; Legako, Dinh, Miller, Adhikari, & Brooks, 2016)

L325 what were the differences in fat levels?

L328 were managed differently in the exploitations – makes no sense in English

L335 Did they look for correlations between scores for each steak between panels?

L344 -363 All of this discussion is all very speculative. Since the US panel didn’t rate liver, how do they even know they are less sensitive? Consumers tend to like what they are used to eating. Meat with 0.85 vs 5.85 % fat content will have very different sensory properties. It does seem a weakness in the study, as I’m certain that livery, fatty and off flavour are concepts that both US and Spanish sensory panels can very easily understand. They are not mysterious terms!

L364-373 & Figure 2:

Why not just do a PCA plot with sensory attributes also shown? This is uninterpretable without the attributes shown in the figure!

L388 highly appreciable? Do they mean high acceptance?

L391 395 The fact that only 6 trained panellists were used in this study is a major problem with this study. Totally underpowered, 6 is the very bare minimum people required! They should have used at least 10 or 12. I am surprised that so few were used, which makes me seriously doubt the experience of these researchers! Why bother paying for samples to be transported to two different countries and use so few panellists? Baffling really!

Corbin, C. H., O'Quinn, T. G., Garmyn, A. J., Legako, J. F., Hunt, M. R., Dinh, T. T. N., Rathmann, R. J., Brooks, J. C., & Miller, M. F. (2015). Sensory evaluation of tender beef strip loin steaks of varying marbling levels and quality treatments. Meat Science, 100(0), 24-31.

Frank, D., Ball, A., Hughes, J., Krishnamurthy, R., Piyasiri, U., Stark, J., Watkins, P., & Warner, R. (2016). Sensory and Flavor Chemistry Characteristics of Australian Beef: Influence of Intramuscular Fat, Feed, and Breed. J Agric Food Chem, 64(21), 4299-4311.

Frank, D., Joo, S.-T., & Warner, R. (2016). Consumer Acceptability of Intramuscular Fat. Korean Journal for Food Science of Animal Resources, 36(6), 699-708.

Frank, D., Kaczmarska, K., Paterson, J., Piyasiri, U., & Warner, R. (2017). Effect of marbling on volatile generation, oral breakdown and in mouth flavor release of grilled beef. Meat Science, 133, 61-68.

HEYMANN, H., MACHADO, B., TORRI, L., & ROBINSON, A. L. (2012). HOW MANY JUDGES SHOULD ONE USE FOR SENSORY DESCRIPTIVE ANALYSIS? , 27(2), 111-122.

Hunt, M. R., Garmyn, A. J., O'Quinn, T. G., Corbin, C. H., Legako, J. F., Rathmann, R. J., Brooks, J. C., & Miller, M. F. (2014). Consumer assessment of beef palatability from four beef muscles from USDA Choice and Select graded carcasses. Meat Science, 98(1), 1-8.

Joo, S. T., Hwang, Y. H., & Frank, D. (2017). Characteristics of Hanwoo cattle and health implications of consuming highly marbled Hanwoo beef. Meat Science, 132, 45-51.

Lee, B., Yoon, S., & Choi, Y. M. (2019). Comparison of marbling fleck characteristics between beef marbling grades and its effect on sensory quality characteristics in high-marbled Hanwoo steer. Meat Science, 152, 109-115.

Legako, J. F., Dinh, T. T. N., Miller, M. F., Adhikari, K., & Brooks, J. C. (2016). Consumer palatability scores, sensory descriptive attributes, and volatile compounds of grilled beef steaks from three USDA Quality Grades. Meat Science, 112, 77-85.

Thompson, J. M. (2004). The effects of marbling on flavour and juiciness scores of cooked beef, after adjusting to a constant tenderness. Australian Journal of Experimental Agriculture, 44(7), 645-652.

Thompson, J. M., Polkinghorne, R., Hwang, I. H., Gee, A. M., Cho, S. H., Park, B. Y., & Lee, J. M. (2008). Beef quality grades as determined by Korean and Australian consumers. Australian Journal of Experimental Agriculture, 48(11), 1380-1386.

Author Response

Dear reviewer #2,

In this document, we address the comments and points raised, and we indicate the changes made in the manuscript (marked using Track Changes). The original comments made by the reviewer appear in italics and blue type, whereas our responses appear in normal case.

Comments and Suggestions for Authors 

Overall, this study has some merit, although the choice of using only 6 trained panellists gives me low confidence in the scientific judgement and experience of the authors. It is the absolute minimum number any self-respecting sensory scientist would use, and that would be just for screening purposes (HEYMANN, MACHADO, TORRI, & ROBINSON, 2012), ttps://www.ifst.org/sites/default/files/Sensory%20and%20Consumer%20Science-.pdf. 

Response: The authors appreciate the feedback and comments. The mentioned reference is related to a wine sensorial tasting with more than 20-30 descriptors; conclusion is valid for this beverage. There should not be extrapolated to other products. Other authors highlight the importance of the panel performance and effectiveness over the panellist number. The greater the number of sessions and the lower the number of attributes, the lower the number of judges needed (Simiqueli et al. 2015 doi:10.1016/j.foodqual.2015.03.019; Pagès & Périnel. 2004; doi:https://doi.org/10.1111/j.1745-459X.2004.tb00148.x and Tomic et al. 2013; doi:10.1016/j.foodqual.2012.06.012). 

A new sentence in section 2.7 lines 215-217 has been added:  

“A rigorous methodology according to Gorraiz et al. (2000) was used to establish the descriptive panels and to select attributes to evaluate the sensory quality of meat [27]”. 

Why conduct an international study and put so few resources into it? I don’t get it! It is a serious potential flaw which needs explanation. Were the panellists highly trained in just tasting beef? How many years’ average of experience? How much training? Paid or unpaid? Much more information required, to convince me that 6 panellists was a wise choice!  

Response: A new sentence in section 2.7 lines 215-218 has been added:  

“A rigorous methodology according to Gorraiz et al. (2000) was used to establish the descriptive panels and to select attributes to evaluate the sensory quality of meat [27]”. Panellists were highly trained in beef tasting with an average of 10 years’ experience. After each project tasting, they are rewarded with a present (voucher for beef) whose value depends on the time devoted to (re-training/harmonizing and tasting. 

In general, the English expression sounds very Spanish. Have the US co-authors had input into fixing up the English expression? I would say not. Have they even seen it? There is a lot of room for improvement! Please thoroughly revise the English using native speakers!

Response: English has been reviewed and improved. 

Abstract 

Please be clear and unambiguous in your descriptions  

Response: Abstract has been enhanced. 

L16 the trained panel assessed taste and texture so remove the word taste 

Response: The word “taste” has been removed. 

L17 It is obvious they are already obtained… remove 

Response: The words “Once obtained” has been removed. 

L16 (Pirenaica, Spain) and (Certified Angus Beef, USA) … need to declare from which country 

Response: Origin of the animals has been written in lines 14 and 15.  

“The sensory differences between Pirenaica beef (Spanish origin) and Certified Angus Beef (CAB) (USA origin) assessed by two trained…”. 

L18 I assume they were all frozen, not just the ones sent to the other country! 

Response: Yes, all of them were frozen. The sentence has been re-written Lines 17-18. 

“Then, samples were half cut and frozen and one of the halves shipped to the other counterpart-testing place.” 

L19 visual marbling

Response: L19. The text has been corrected from “visually marbling” to “visual marbling” 

L20 Indeed is not required 

Response: L20. The word “Indeed” has been deleted.

The sentence now begins as” To assess the physicochemical parameters…”. 

L22 greater à higher 

Response: L23. The text has been corrected from “greater” to “higher” 

L32 because of 

Response: The text has been corrected from “because” to “because of” 

L37-42 For example, they can mention the perception of marbling or fat in meat is considered quite differently in different countries (Damian Frank, Joo, & Warner, 2016; Lee, Yoon, & Choi, 2019) 

Response: Due to polishing in English, lines 38-43 have been removed and the following paragraph added:“Besides, consumption habits and preferences may influence clearly the assigned score to meat coming from different origins. Hence, country of origin and production practices can affect both meat quality and consumer acceptance [3,4], as reported by [5], who observed that low or trace marbled beef was acceptable for many Chilean consumers, whereas in Japan and Korea consumers tend to prefer evenly distributed marbling in beef [6]”.

L45 ‘ not required, these regulations establish.. 

Response: Sentence deleted L44.  

L45-59 Much of this information is not necessary, please reduce 

Response: Paragraph deleted L49-56 

L66-77 They really need to describe the very strong relationships between marbling and flavour and tenderness. It is very well established and holds true regardless of country. (D. Frank, Ball, Hughes, Krishnamurthy, Piyasiri, Stark, et al., 2016; Damian Frank, Kaczmarska, Paterson, Piyasiri, & Warner, 2017; Joo, Hwang, & Frank, 2017) 

Response: A new paragraph describing the reviewer recommendation has been added in the manuscript in L62-65 

“The association of marbling degree with juiciness and tenderness of meat is well stablished. Damian Frank et al (2016) showed positive associations between IMF and overall flavour impact as well as juiciness and tenderness for Australian beef [11]. Fat makes the matrix soft and easier to chew therefore influencing meat tenderness [12-14].” 

Figure 1. Surely all the steak samples were frozen. Otherwise the tenderness and flavour will be totally different! 

Response: Figure 1 has been modified to cover the cut in two halves and the frozen stripes before delivering.  

L214 Human ethics approval number in US and Spain? 

Response: Information sheets were provided to each panelist and consents forms signed by them, but the US and Spain approval numbers are not available. 

L216 in both evaluation centres 

Response: The text has been corrected in L215 from “in both analytical places” to “in both evaluation centres” 

L219 Grill temperature, cooking time? 

Response: Sentence in lines 219 and 223 has been clarified in the following way:  

“Steaks were cooked in a 180ºC preheated open-hearth electric grill model 450 (Farberware Cookware, Fairfield, CA, USA) and an open heart-electric grill model GR3000 (JATA S.A, Tudela, Spain) inside aluminium foil turning the steak every 4 minutes till reaching an internal temperature of 70 °C”. 

L221 1 cm 3 is a very tiny amount to taste! 1 g of beef? How could you even chew it? Surely not!? 

Response: The mistake in size has been corrected. Sentence in lines 223-225 is written as: 

“Every steak was then trimmed of any external connective tissue, cut into 3.5x1x1 cm3 rectangle samples, wrapped in coded aluminum foil and kept in a warmer, provided with sand, until analysis

L223 In hot sand? What does that mean? 

Response: Sentence has been clarified as answered in previous comment. 

Are they saying that each panellist tasted 80 pieces of steak? Over 7 sessions? Please be clear how steaks from 40, 20 Angus and 40, 20 Pirenaica loins were allocated? 

Response: In sentence in L228-229 the word “seven judgments” has been changed to “six judgments”. 

“Seven tasting sessions were conducted to obtain at least six judgments per steak.   

Response: A new sentence in section 2.7 has been added L229-230. 

“Briefly, on each tasting session 3 Pirenaica and 3 CBA samples were presented to each judge, except on the first session where two samples per breed were tasted.” 

L244 Marbling scores had a large SD. It is not obvious what the word and superscript number means in Table 1. The whole point of a marbling score is a number. Please just use numbers in table. Interpretation of the numerical scores can be assisted by a table legend. 

Response: Table 1 have been modified accordingly.

Tables: tested in USA, tested in Spain (not at)

Response: L256, L260 and L264. The text has been corrected from “at” to “in”. 

Not sure why a discussion about SD is required in a separate section? Just discuss with data. 

Response: Thanks for the comment. However, the authors consider that including independent sentences about SD on each data sections will be more confusing than explaining all of them together. This is related with the adequacy of the tests performed.   

L271 Pirenaica loins had larger average LEA than CAB à not showed (many other examples of this!)

Response: Sentence added in L278-279

“LEA values of 63.6 and 48.61 cm2 were reported by Piedrafita et al. (2003) and by Panea et al. (2008) for Pirenaica breed [31, 32].”

274 Surely, they have the individual carcass weights for each animal. Why not present the actual correlations with a regression equation? 

Response: a regression equation between LEA and slaughter weight would be highly valuable however all the needed data are not available.

L288 were close to the ones obtained in this study 

Response: L292-293. Sentence “Both values, for Pirenaica and for CAB, were closed to the ones obtained on this study” has been deleted and a new sentence has been added:  “Both data support the values obtained in this study (5.9 kg and 3.86 kg for Pirenaica and CAB breeds, respectively).” 

L293 Unless they performed a correlation (they have the data so why not?) please do not discuss it. If you do please show r value, p value and regression equation and p value. 

9-fold 

Response: Sentence L297-299 has been rewritten:  

“Mean fat content for CAB samples was higher (p < 0.0001) than that of Spanish striploins, which correlates well with the statistical differences found in marbling score (Table 1).” 

L297 Please cite other work like (D. Frank, et al., 2016; Thompson, 2004; Thompson, Polkinghorne, Hwang, Gee, Cho, Park, et al., 2008) that also show this well-known relationship between fat and moisture. There are many studies showing this. 

Response: The mentioned references have been added in L302 of the manuscript. 

L313 Why state UIUC? Just Say the US sensory panellists 

Response: The text has been corrected in L317 from “In UIUC, trained USA sensory panelists” to “The US sensory panelist both evaluation centres” 

Please discuss the well-known relationship between marbling level and sensory attributes, including flavour release regardless of breed (Corbin, O'Quinn, Garmyn, Legako, Hunt, Dinh, et al., 2015; Damian Frank, Kaczmarska, Paterson, Piyasiri, & Warner, 2017; Hunt, Garmyn, O'Quinn, Corbin, Legako, Rathmann, et al., 2014; Legako, Dinh, Miller, Adhikari, & Brooks, 2016) 

Response: L327-333. Paragraph added.

“It is well-known the relationship between marbling level and sensory attributes, including flavour perception regardless of breed. In such a way that a higher intramuscular fat percentage is clearly linked to overall liking [40,41-43]. Fat level is the primary driver of beef flavor acceptability justifying the results showed in this work in which irrespective of the panellists country of origin, the CAB beef (fat= 5.84%) was perceived as more tender and having an increased beef flavour compared to Pirenaica beef (fat= 0.85%).”

L325 what were the differences in fat levels? 

Response: L337-338 Sentence added.

“The study reported fat levels of 8.82% and 6.12% for the US and Australian beefs, respectively.” 

L328 were managed differently in the exploitations – makes no sense in English

Response: L-338-340 Sentence re-written.

“A different study reported that US consumers did not detect differences in beef flavor when tasting meat from different breeds although they were produced under similar livestock practices” 

L335 Did they look for correlations between scores for each steak between panels? 

Response: The correlation in between individual steask´s scores and panels were not considered. However, the manuscript focuses the comparison between both panels by performing the discriminant analysis were it is demonstrated that both panels shown the same trend in the attributes evaluation.

L344 -363 All of this discussion is all very speculative. Since the US panel didn’t rate liver, how do they even know they are less sensitive? Consumers tend to like what they are used to eating. Meat with 0.85 vs 5.85 % fat content will have very different sensory properties. It does seem a weakness in the study, as I’m certain that livery, fatty and off flavour are concepts that both US and Spanish sensory panels can very easily understand. They are not mysterious terms! 

Response: L362-366 develops the lines 344-363 in order to provide further explanation 

“For USA consumers, an intramuscular fat level below 3% is not acceptable. As indicated by Muhammad Sajid Arshad et al. (2018) [45] several are the factors influencing beef flavor with special Response reference to fatty acids. This, together with the mineral composition and the individual fatty acid profile affects meat off-flavors such as liver development [46].”  

L364-373 & Figure 2: 

Why not just do a PCA plot with sensory attributes also shown? This is uninterpretable without the attributes shown in the figure! 

Response: PCA is a technique focused on variables and these can be grouped to generate a reduced number of new variables. In the present research, there are only 3 attributes common to both laboratories. This number is insufficient to obtain new “groups of variables”. On the contrary, DA is a technique focused on the samples and they can be grouped according to the variables that differentiate them the most. To facilitate the interpretation of figure 2, the following caption is included at the bottom of graphic:  

Function 1 (texture factor) was associated with tenderness and juiciness, and function 2 (flavor factor) was associated with beef flavor. 

L388 highly appreciable? Do they mean high acceptance? 

Response: L405-406. The sentence has been re-written.

From “In terms of sensorial quality, the CAB striploins were considered as highly appreciable” to “In terms of sensory quality, the panels in the two countries concur that CAB striploins outperformed Pirenaica samples in juiciness, tenderness and flavor.” 

 L391 395 The fact that only 6 trained panellists were used in this study is a major problem with this study. Totally underpowered, 6 is the very bare minimum people required! They should have used at least 10 or 12. I am surprised that so few were used, which makes me seriously doubt the experience of these researchers! Why bother paying for samples to be transported to two different countries and use so few panellists? Baffling really! 

Response: The authors appreciate the feedback and comment, The above reference indicated by the reviewer is related to a wine sensorial tasting with more than 20-30 descriptors; conclusion is valid for this beverage. There should not be extrapolated to other products. Other authors highlight the importance of the panel performance and effectiveness than the panellist number. The greater the number of sessions and the lower the number of attributes, the lower the number of judges (Simiqueli et al. 2015  doi:10.1016/j.foodqual.2015.03.019; Pagès & Périnel. 2004; doi:https://doi.org/10.1111/j.1745-459X.2004.tb00148.x and Tomic et al. 2013; doi:10.1016/j.foodqual.2012.06.012). 

A new sentence in section 2.7 lines 215-218 has been added:  

A rigorous methodology according Gorraiz et al. (2000) was used to establish the descriptive panels and to select attributes to evaluate the sensory quality of meat. 

 Corbin, C. H., O'Quinn, T. G., Garmyn, A. J., Legako, J. F., Hunt, M. R., Dinh, T. T. N., Rathmann, R. J., Brooks, J. C., & Miller, M. F. (2015). Sensory evaluation of tender beef strip loin steaks of varying marbling levels and quality treatments. Meat Science, 100(0), 24-31. 

Frank, D., Ball, A., Hughes, J., Krishnamurthy, R., Piyasiri, U., Stark, J., Watkins, P., & Warner, R. (2016). Sensory and Flavor Chemistry Characteristics of Australian Beef: Influence of Intramuscular Fat, Feed, and Breed. J Agric Food Chem, 64(21), 4299-4311. 

Frank, D., Joo, S.-T., & Warner, R. (2016). Consumer Acceptability of Intramuscular Fat. Korean Journal for Food Science of Animal Resources, 36(6), 699-708. 

Frank, D., Kaczmarska, K., Paterson, J., Piyasiri, U., & Warner, R. (2017). Effect of marbling on volatile generation, oral breakdown and in mouth flavor release of grilled beef. Meat Science, 133, 61-68. 

HEYMANN, H., MACHADO, B., TORRI, L., & ROBINSON, A. L. (2012). HOW MANY JUDGES SHOULD ONE USE FOR SENSORY DESCRIPTIVE ANALYSIS? , 27(2), 111-122. 

Hunt, M. R., Garmyn, A. J., O'Quinn, T. G., Corbin, C. H., Legako, J. F., Rathmann, R. J., Brooks, J. C., & Miller, M. F. (2014). Consumer assessment of beef palatability from four beef muscles from USDA Choice and Select graded carcasses. Meat Science, 98(1), 1-8. 

Joo, S. T., Hwang, Y. H., & Frank, D. (2017). Characteristics of Hanwoo cattle and health implications of consuming highly marbled Hanwoo beef. Meat Science, 132, 45-51. 

Lee, B., Yoon, S., & Choi, Y. M. (2019). Comparison of marbling fleck characteristics between beef marbling grades and its effect on sensory quality characteristics in high-marbled Hanwoo steer. Meat Science, 152, 109-115. 

Legako, J. F., Dinh, T. T. N., Miller, M. F., Adhikari, K., & Brooks, J. C. (2016). Consumer palatability scores, sensory descriptive attributes, and volatile compounds of grilled beef steaks from three USDA Quality Grades. Meat Science, 112, 77-85. 

Thompson, J. M. (2004). The effects of marbling on flavour and juiciness scores of cooked beef, after adjusting to a constant tenderness. Australian Journal of Experimental Agriculture, 44(7), 645-652. 

Thompson, J. M., Polkinghorne, R., Hwang, I. H., Gee, A. M., Cho, S. H., Park, B. Y., & Lee, J. M. (2008). Beef quality grades as determined by Korean and Australian consumers. Australian Journal of Experimental Agriculture, 48(11), 1380-1386. 

Response: The authors thank the reviewer for the references provided. The mentioned references have been reviewed and several of them have been included in the manuscript where considered adequate. 

Additionally, the authors wish to indicate to the reviewer that due to the removal, modification or addition of some paragraph, the references have been updated. However, no change control for this action was kept due to the difficulties this action provided on reviewing the new version of the manuscript.

Reviewer 3 Report

This study dealt with comparison of the sensory properties between two breeds of beef by panelists from different geographically and culturally different countries. Even if the panelists do training, it is difficult to expect the same results as it is highly dependent on the panelist' food-cultural experience. Nevertheless, in this study, I think that the experimental design was suitable for the purpose, the results were well derived, and the discussion was well done. I think that this manuscript was well written, making it easy for readers to understand and useful. However, the purpose of this study is not sufficiently presented in the Abstract section.

Author Response

Dear reviewer #3,

In this document, we address the comments and points raised, and we indicate the changes made in the manuscript (marked using Track Changes). The original comments made by the reviewer appear in italics and blue type, whereas our responses appear in normal case.

Comments and Suggestions for Authors 

Comments and Suggestions for Authors 

This study dealt with comparison of the sensory properties between two breeds of beef by panelists from different geographically and culturally different countries. Even if the panelists do training, it is difficult to expect the same results as it is highly dependent on the panelist' food-cultural experience. Nevertheless, in this study, I think that the experimental design was suitable for the purpose, the results were well derived, and the discussion was well done. I think that this manuscript was well written, making it easy for readers to understand and useful. However, the purpose of this study is not sufficiently presented in the Abstract section. 

Response: The authors appreciate the feedback and comments. 

Abstract has been re-written in order to better present the study purposes.

L87-90 has been rewritten in order to provide a better understanding of the manuscript objectives.

“The objective of this study was to assess differences on physicochemical parameters of two different biological types of cattle, Pirenaica and Certified Angus Beef (CAB). Besides, the effect of the country of origin on the beef sensory attributes was evaluated by two trained panels in Spain and USA.” 

Additionally, the authors wish to indicate to the reviewer that due to the removal, modification or addition of some paragraph, the references have been updated. However, no change control for this action was kept due to the difficulties this action provided on reviewing the new version of the manuscript.

Round 2

Reviewer 2 Report

It appears that the authors have addressed suggestions adequately. Good job!

There a few spelling/English issues (I don't have time to address) but I urge the proofreader to make appropriate changes. 

Author Response

Moderate English changes required have been done in the manuscript. The authors have done another changes in order to improve the paper according to the editor´s recommendations.